# Research on the evaluation of China's Supply Chain Finance policy based on text mining

**Mingyang Li**[1]*, **Yin Dong**[2]

1 School of Software, Harbin Institute of Information Technology, Harbin, Heilongjiang, China, 2 School of Economics and Management, Hefei University, Hefei, Anhui, China

* limoyvez@gmail.com

## Abstract

Supply Chain Finance (SCF) aims to manage the capital flow, logistics flow, and information flow of small and medium-sized enterprises (SMEs) in the upstream and downstream of the supply chain while optimizing supply chain risk control. Like other types of financial services, the development of SCF is highly influenced by policy factors; however, related research remains relatively limited. This study aims to explore the current state of SCF policymaking in China and provide scientific recommendations for the development of SCF from a policy perspective. First, using the BERTopic model, 3,439 SCF-related academic papers and 181 central-level SCF policy texts from the CNKI database were analyzed for thematic clustering. Then, by comprehensively considering the thematic distribution of SCF research and the operational characteristics of SCF, the Policy Modeling Consistency (PMC) Index model was constructed to evaluate SCF policy texts. The findings reveal several issues in China's SCF policymaking: limited thematic focus, fluctuating levels of policy formulation, and significant homogenization of policy content. The study proposes several optimization recommendations for SCF policies, including expanding the scope of policy focus, fostering synergy among different types of policies, diversifying the use of policy tools, and broadening the range of target groups addressed by policies.

## Introduction

### Background

SMEs play a crucial role in China's economy, serving as the foundation for the orderly progression of social and economic activities. However, existing research highlights that financing difficulties remain one of the primary challenges for SMEs in their operations and production. These challenges are largely driven by the high costs associated with technological upgrades and transformations, as well as the complexities of credit transmission [1,2]. SCF, as an innovative financing model that integrates transaction flows, logistics, information flows, and capital flows, seeks to optimize cash flow management across supply chains. By addressing information asymmetry and mitigating credit risks, SCF provides an effective solution to the persistent issues of financing difficulties and high costs [3].

The development of SCF is a multifaceted and systematic process, driven by the integration of digital technologies, the evolution of business models, and the gradual refinement of

**Data availability statement:** The data underlying this study is available on the Kaggle repository at the following link: (https://kaggle.com/datasets/98a5f29f8ce28f7a4c8d222b0c81b-7077889ca71fc4d337847e74d27cf0d0602).

**Funding:** This study did not receive any funding.

institutional frameworks, supply chain ecosystems, and financial systems. At the technological level, existing research focuses on fostering the integration of technology and SCF by developing new technologies and exploring their applications in SCF, thereby promoting its innovative growth [4]. At the social level, scholars have examined the relationships between supply chain participants, emphasizing that collaboration and synergy among these participants are critical for the advancement of SCF [5]. From a business perspective, SCF is a financing activity that involves multiple stakeholders within the supply chain and leverages information technology. It combines the characteristics of both financial services and financial technology [6,7]. China's 20th National Congress underscored the importance of SCF as a tool to promote real economic development, strengthen supply chain resilience, and support initiatives such as "rural revitalization." This has established a clear direction for the sustainable development of SCF services at the macro level.

Compared to developed countries, China has a relatively short history of SCF development. Over the past two decades, however, SCF has experienced rapid growth. In 2018, under the exploration of industrial-driven initiatives, a new SCF model—characterized by industrial internet platforms and financial technology applications—emerged, marking the beginning of a new phase of diverse development. From a policy perspective, the release of the "Guiding Opinions on Actively Promoting Supply Chain Innovation and Application" in October 2017 was the first major policy explicitly aimed at advancing SCF innovation and platform development. Subsequent policies, such as the "Guiding Opinions on Promoting SCF to Serve the Real Economy" (July 2019) and the "Opinions on Standardizing SCF Development to Support the Stabilization and Upgrading of Supply Chain Industry Chains," provided regulatory frameworks to guide SCF's growth from an institutional standpoint.

Considering the evolution of business models and policy development, China's SCF development can be categorized into three stages: the market observation stage (Stage-1) from 2015 to 2018, the guided innovation stage (Stage-2) from 2018 to 2020, and the standardized development stage (Stage-3) from 2020 to the present.

China's rapid SCF development is expected to continuously refine and enhance the associated policy framework. Increasing attention from SCF participants and policymakers has spurred relevant research, the findings of which significantly influence policy optimization and regional economic growth by reducing costs and improving efficiency. By analyzing central government policies at different stages of SCF development, we can better understand the government's priorities and policy trends. This study leverages the BERTopic model to analyze Chinese SCF policies and research abstracts while constructing a PMC index evaluation model. The goal is to assess the current state of China's SCF policy system at the central level and provide scientific insights for improving SCF policymaking.

## Literature review

### SCF policy-related research

SCF has played a significant role in alleviating SMEs' financing difficulties and enhancing the stability of supply chains [8,9]. Research on SCF from a policy perspective primarily focuses on two areas: policy formulation recommendations and policy evaluation. Existing studies emphasize the role of SCF policies in promoting financial innovation, improving supply chain efficiency, and supporting industrial upgrading [10–12]. For example, some studies have analyzed the impact of government-issued guidelines on SCF, highlighting their role in fostering collaboration among financial institutions, core enterprises, and SMEs, as well as exploring how SCF policies address credit constraints and encourage technological innovations such as blockchain and digital platforms [13,14].

However, there are clear deficiencies in research focusing on SCF policies. First, the thematic priorities of SCF policies remain underexplored, with limited attention paid to analyzing the diversity and evolution of policy objectives over time. The lack of a systematic framework for comparing policy content across different periods or regions further hinders a comprehensive understanding of the impact of SCF policies [15]. Second, most studies rely on qualitative methods or descriptive analyses, lacking the rigor and replicability offered by quantitative methods. Few studies have employed advanced text mining techniques to systematically analyze SCF policy content or evaluate their effectiveness using quantitative models [16]. While some studies have assessed the consistency of SCF policies with broader economic goals, such as rural revitalization and supply chain resilience, there has been insufficient exploration of how these policies address specific challenges, such as policy homogenization or over-reliance on core enterprises [17–19].

## SCF Development

When we combine the pertinent studies on fostering SCF development, we discover that the current research viewpoints concentrate on the following two aspects:

One of them is to empower SCF development through technological innovation. For instance, supply chain finance businesses adopt digitalization and blockchain technology to enhance their transparency, security, and efficiency [20]. Research in this field mostly focuses on the application effect of blockchain-based smart contracts and secure transaction platforms in simplifying financial processes [21]. Meanwhile, building a collaboration platform and establishing an SCF ecosystem based on digital technology is another important direction for SCF development research. Research has extensively demonstrated that forming partnerships among diverse stakeholders, including banks, fintech companies, and supply chain participants, can foster synergies and enhance SMEs' financial access [22,23]. Furthermore, improving risk management and credit assessment systems based on information technology innovation is a hot research topic in the field of SCF. Effective risk management practices and advanced credit assessment models have always been crucial to steadily improving the level of SCF business development. Most of the studies in this part emphasize the importance of low-latency data processing, machine learning algorithms, and predictive models in assessing creditworthiness and reducing supply chain financing risks [24,25].

The second is to propose policy recommendations to support the development of SCF based on research in SCF-related areas. These policy recommendations mostly focus on the development of the regulatory system, the improvement of incentives, the enhancement of service capacity, and talent cultivation [26–29]. The development of SCF in China faces significant challenges due to the extremely limited existing research and the absence of a relevant research system.

Using a quantitative analysis of SCF policies, this paper carefully looks at the level of relevant policy formulation at the central level to help SCF businesses grow. The goal is to set the stage for future SCF research and policy formulation.

## Policy evaluation methods

Policy evaluation plays a critical role in determining the effectiveness, consistency, and alignment of policies with their intended objectives [30]. Traditional policy evaluation methods can be categorized into three types: qualitative, quantitative, and hybrid methods.

Qualitative methods (e.g., expert judgment, the Delphi technique, and case studies) are widely used due to their flexibility in interpreting complex policy contexts [31–33]. However,

these methods are often criticized for their subjectivity, potential biases, and lack of reproducibility [34].

Quantitative methods (e.g., Analytic Hierarchy Process (AHP), Social Network Analysis (SNA), Backpropagation Neural Network (BPNN), and Multi-Attribute Decision Analysis (MADA)) aim to enhance objectivity by introducing measurable variables and data-driven techniques [35–38]. Nonetheless, these methods face several challenges. For instance, the AHP method is prone to subjective bias when assigning weights to evaluation indicators, while the SNA method often lacks the semantic depth required to fully capture the content of policy texts [39,40]. Similarly, the BPNN method suffers from issues such as overfitting and sensitivity to hyperparameter settings, which can diminish the reliability of the results [41]. Although the MADA method is effective in multi-criteria decision-making, it struggles to fairly allocate weights among influencing factors, thereby affecting its credibility [42].

In contrast, the PMC Index model addresses some of these limitations by providing a more comprehensive and systematic evaluation framework [43]. By incorporating a wider range of variables and employing a structured approach to weight assignment, the model minimizes the subjectivity of policy evaluation. Furthermore, its multidimensional evaluation framework considers various policy attributes, such as the nature, scope, objectives, and effectiveness of policies, significantly improving the accuracy of policy evaluation [44].

Although the PMC Index model offers a robust framework for policy evaluation, the validity of its results depends heavily on the accurate identification of variables and indicators that reflect the essence of the policies being evaluated. To improve the objectivity and accuracy of indicator selection, an increasing number of studies have attempted to use deep learning technologies to enhance the credibility of indicator selection [45]. The BERTopic model, leveraging BERT's contextualized language representations, performs thematic clustering and semantic analysis on unstructured text data.The application of this model in constructing evaluation indicator systems presents several key advantages. First, it helps systematically identify text topics and priorities, ensuring that the selected indicators reflect the actual focus of the policies under study [46]. Second, by clustering themes in policy and research texts, the BERTopic model identifies the distribution, overlap, and redundancy of topics in related research fields, providing a data-driven foundation for improving the evaluation framework [47]. Third, its ability to process large-scale datasets enhances the comprehensiveness and reliability of the evaluation indicator system [48].

This study integrates the BERTopic model with the PMC Index model to enhance the effectiveness of SCF policy evaluation indicator selection, thereby improving the accuracy of SCF policy evaluation results.

## Concept definition

This study spans the fields of financial services, computer technology, and policy research, making it highly interdisciplinary. To improve the clarity and readability of the research, it is necessary to define the relevant technical terms used in the study. The definitions of these terms and their interpretations within this study are detailed in Table 1.

## Research framework

The structure of this study is as follows: Part II analyzes the text clustering analysis of Chinese SCF policy samples and SCF research paper abstracts based on the BERTopic model and constructs the PMC-Index model to quantitatively evaluate SCF policies. The third part forms the empirical results and draws the PMC-Surface diagram; the fourth part synthesizes the results of text clustering analysis and PMC index evaluation of SCF policies to summarize the full text

**Table 1. Concept definition of research terminology.**

| Concept | Definition | Interpretation methods in this study |
|---|---|---|
| SCF | SCF refers to a set of financial practices and solutions aimed at optimizing the flow of capital, logistics, and information across the supply chain. It facilitates financing for supply chain participants by leveraging the creditworthiness of core enterprises and integrating processes such as receivables financing, inventory financing, and prepayment financing. | Same as definition |
| SCF policy | Policy documents focusing on supply chain finance and its development | Policy text that mentions SCF in policy clauses and clearly expresses the attitude towards developing or adopting SCF |
| Text Mining | Text mining is the process of extracting meaningful information, patterns, and knowledge from unstructured textual data using techniques such as natural language processing, machine learning, and statistical analysis. | Same as definition |
| Topic Clustering | Topic clustering is a text mining technique used to group textual data into clusters of related themes or topics. It identifies latent semantic structures within a corpus and organizes documents accordingly, helping to uncover meaningful patterns and trends. | Same as definition |

and put forward suggestions for policy optimization and adjustment. The research framework of the study is shown in Fig 1.

## Data and model

### Data sources

This study conducts a comprehensive combing of policies that promote the development of SCF and obtains SCF policy texts used for quantitative analysis in this paper by searching publicly available documents and materials on the websites of the State Council and the relevant ministries and commissions under it, as well as Internet platforms such as Beida Faber [49]. The following criteria guided the evaluation of the gathered texts. First, we selected the SCF policy texts that China implemented before January 1, 2024, and are currently in effect. Furthermore, the State Council and pertinent ministries are the primary entities responsible for policy formation. We classify the policy documents based on their content, selecting only those that are crucial for the development of SCF and explicitly endorse or implement steps to foster its growth. Specifically, the chosen papers encompassed laws, regulations, views, measures, notifications, announcements, and other documents that directly conveyed policy goals. However, we excluded letters, answers, government work reports, and leaders' speeches from our consideration. Based on this principle, this study collects 181 valid policy samples.

When making the PMC indicator system, we think about both how complete and fair the variable selection is, as well as the regional differences seen in policy evaluation studies [50]. In this study, we utilized a Python-based crawler program to extract the titles, abstracts, keywords, and additional details of 3,439 Chinese academic papers from the CNKI database, using the keyword "SCF development."

### Main technologies for indicator selection

The core purpose of thematic clustering is to uncover latent themes or patterns within a corpus of text, enabling researchers to organize and interpret textual information more

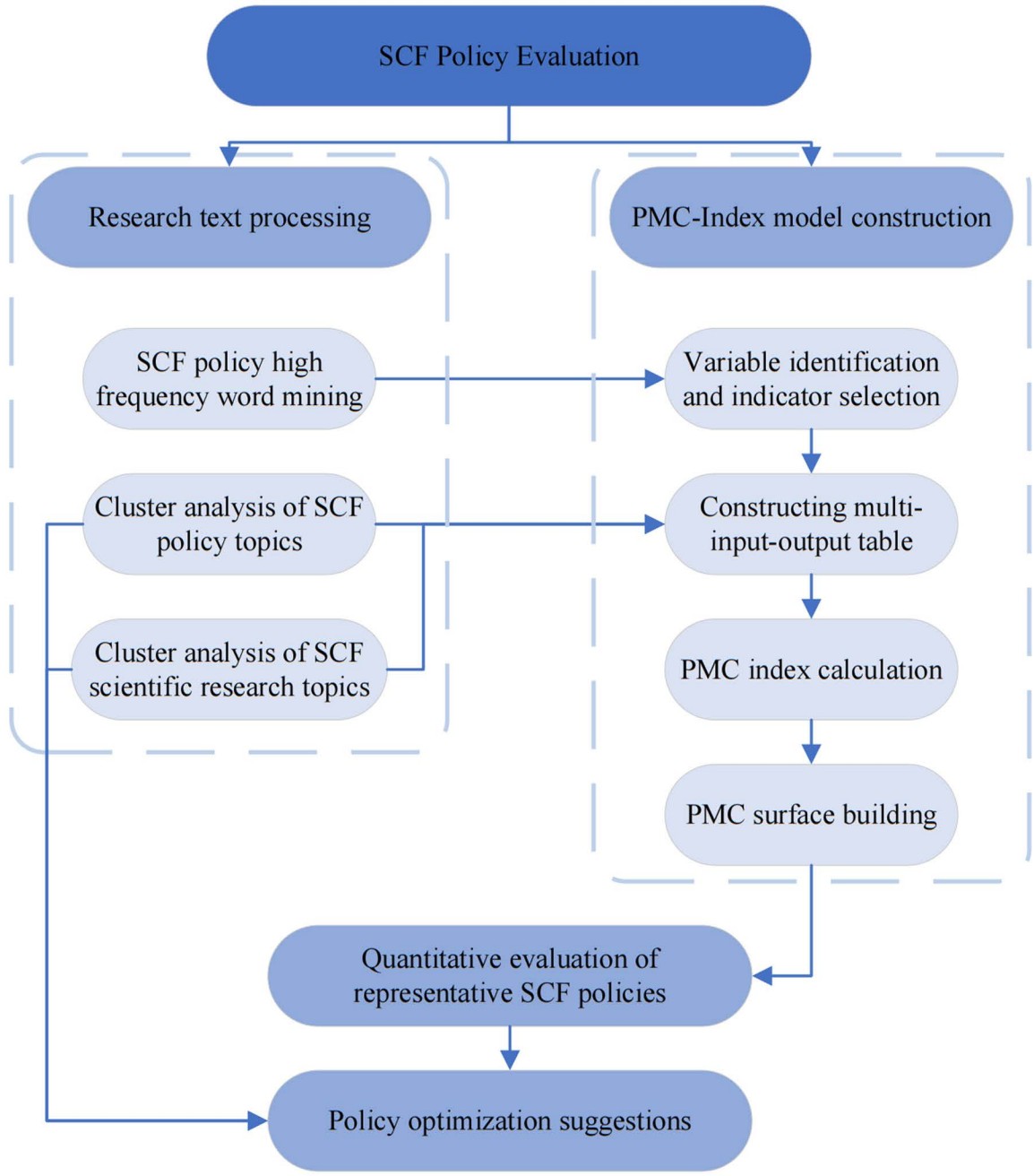

**Fig 1. Research framework.**

effectively. Traditional methods, such as Latent Dirichlet Allocation (LDA) and Non-Negative Matrix Factorization (NMF), have been widely applied in this field. However, these methods rely on bag-of-words representations, which fail to capture the semantic and contextual relationships between words, leading to the loss of critical information. Additionally, these methods often require predefining the number of topics, which introduces subjectivity and may reduce the accuracy of clustering results.

In recent years, transformer-based models, exemplified by BERT (Bidirectional Encoder Representations from Transformers), have significantly improved the accuracy and depth of textual analysis. Built on BERT, the BERTopic model represents a major breakthrough in thematic clustering analysis [51]. By utilizing BERT's deep contextual embeddings, the BERTopic model captures semantic relationships between words and phrases, enabling more precise topic identification. Unlike traditional models, BERTopic also integrates dimensionality reduction techniques such as Uniform Manifold Approximation and Projection (UMAP) and clustering algorithms like Hierarchical Density-Based Spatial Clustering of Applications with Noise (HDBSCAN), which automatically determine the optimal number of topics. This makes it particularly suitable for analyzing complex and diverse datasets. In this study, the application of the BERTopic model enables more efficient and comprehensive analysis of SCF-related texts, providing a solid foundation for constructing a PMC Index-based policy evaluation framework.

## SCF policy and academic research text mining

Policy texts and academic research papers contain a large amount of information. This section systematically combs through the research results related to China's SCF policy and SCF development based on the BERTopic model, and applies the text mining method of thematic clustering to obtain information on the current status of China's SCF development research, business model characteristics, etc., with the aim of accurately identifying the variables of the evaluation model, and selecting the indicators for analysis.

## SCF policy high frequency words

We manually processed the policy samples to obtain a total of 255 policy terms related to SCF. After cutting the words based on the Jieba Chinese thesaurus, we first analyzed the word frequency statistics. We filtered out words with low information content and those that did not reflect the policy purpose, such as "carry out," "provide," "promote," etc., and added them to the deactivated word list [52]. Table 2 lists the top 30 keywords with the highest frequency of occurrence.

Table 2. Statistics of high-frequency words in China's SCF policy.

| Number | High-frequency words | Frequency of occurrence | Number | High-frequency words | Frequency of occurrence |
| --- | --- | --- | --- | --- | --- |
| 1 | Supply Chain | 334 | 16 | Industry Chain | 49 |
| 2 | Finance | 208 | 17 | Risks | 47 |
| 3 | Enterprise | 164 | 18 | Information | 46 |
| 4 | Financing | 150 | 19 | Upstream and Downstream | 44 |
| 5 | Support | 93 | 20 | Bills | 41 |
| 6 | Development | 86 | 21 | Business | 40 |
| 7 | Services | 73 | 22 | Micro and Small Business | 37 |
| 8 | Financial Institutions | 71 | 23 | Pledge | 37 |
| 9 | Financial Services | 70 | 24 | Logistics | 35 |
| 10 | Core | 65 | 25 | Management | 34 |
| 11 | Encouragement | 60 | 26 | Transactions | 33 |
| 12 | Receivables | 54 | 27 | Cooperation | 32 |
| 13 | Products | 53 | 28 | Organizations | 31 |
| 14 | Innovation | 52 | 29 | Data | 31 |
| 15 | Platforms | 52 | 30 | Credits | 30 |

Specifically, the frequent use of terms such as "supply chain," "enterprise," and "core" highlights the government's strong emphasis on the role of SCF in core supply chain enterprises and other participants. Similarly, the prominence of terms like "finance," "financing," "support," "financial institutions," and "financial services" underscores the central government's focus on the financial activities of supply chain enterprises, reflecting the importance of SCF development in China. Furthermore, the appearance of words such as "encourage," "product," "innovation," and "platform" indicates the government's commitment to fostering innovation and development in SCF and related financial products. Terms like "industry chain," "risk," "information," and "upstream and downstream" signal the government's attention to addressing credit issues in SCF development, with a strong emphasis on enhancing downstream credit systems for SCF enterprises.

## Cluster analysis of SCF policies and scientific research topics

The selection of variables and parameter settings is critical to constructing the PMC model. This study conducts thematic clustering of policy and research samples to create a clustering diagram that visually represents the key focus areas of China's SCF policies and research. The diagram highlights the alignment between policy priorities and research objectives, ensuring the PMC model's evaluation index system is both impartial and effective.

We conducted thematic clustering of the research text using the BERTopic model. The Jieba library served as the basis for the word-cutting process, while the Sentence Transformers model of the Hugging Face platform generated the word vectors. Finally, we drew a thematic clustering map using DataMapPlot, which is a small library [53,54]. Fig 2 shows how the topics of 255 Chinese articles about central-level SCF policy are grouped, and Fig 3 shows how the topics are spread out in the abstracts of 3439 highly relevant Chinese academic papers to the development of SCF. In order to enhance the presentation of information, we have

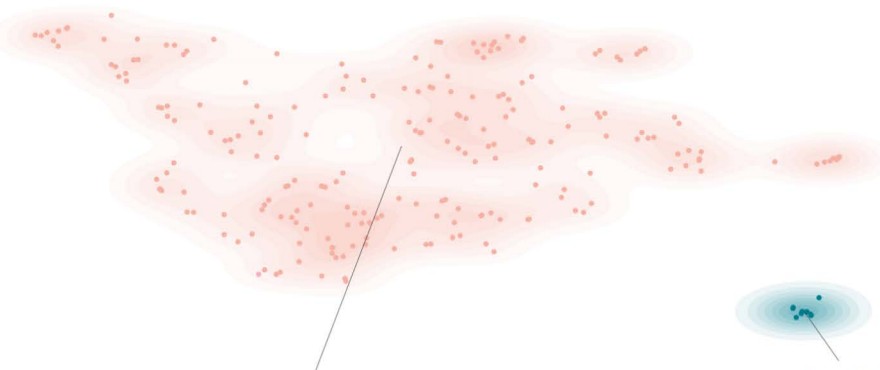

**Fig 2. SCF policy theme clustering.**

substituted the text clustering themes in Fig 3 with numerical representations. For further details, please refer to Table 3.

Analysis of Fig 2 reveals that China's central-level policies primarily focus on the relationship between SCF and enterprise financing. Secondary priorities include SCF's role in agricultural issues and farmers' returns, followed by the application of technological tools such as blockchain. Overall, these policies aim to advance SCF as a means to support corporate financing.

The primary focus of Chinese-language research on the growth of SCF in China is on several areas, as evidenced by a comprehensive examination of Fig 3 and Table 3.

(1) The keywords of the eight topics identified by the serial numbers "29-14-8-9-43-10-7-17" include "SMEs," "financing," "constraints," "innovation," and "information." This section examines the effects of the COVID-19 pandemic and economic development constraints, aligning with the core ideas of the policy theme Topic-0. Both policy and academic research demonstrate a shared interest in optimizing SCF to improve funding access for small businesses.

(2) The keywords of the eight topics with serial numbers "36-19-41-31-12-11-44-34" include "agriculture," "farmers," "revitalization," and "poverty." These topics correspond to the

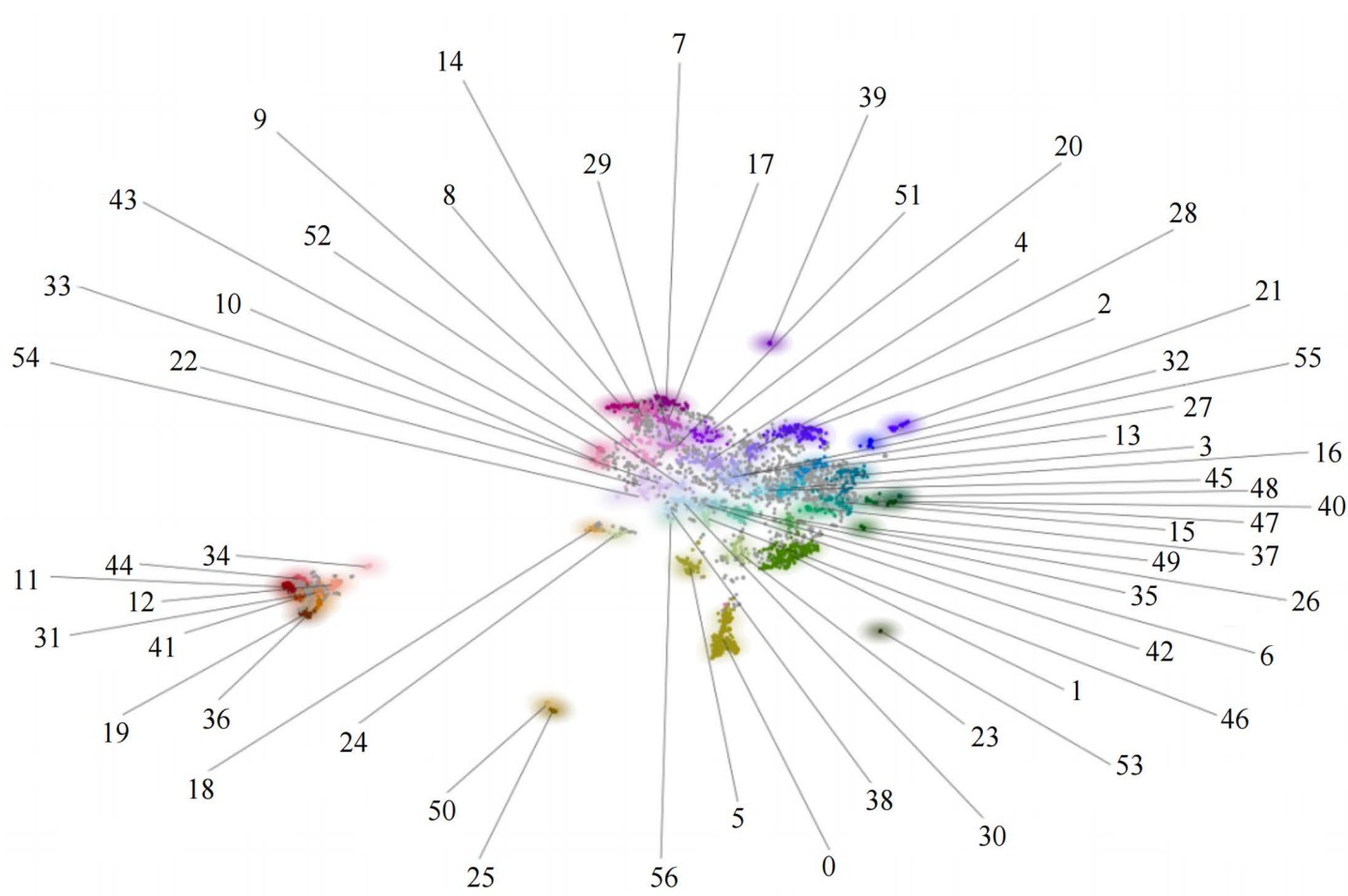

**Fig 3. Cluster distribution of SCF research topics in China.**

**Table 3. Comparison of SCF Research Topics.**

| Topic 0-28 | Topic 29-56 |
|---|---|
| 0 Blockchain_Technology_Application | 29 Epidemic_Impact_Economy |
| 1 Internet_Online_E-commerce | 30 Shenzhen Development Bank_Business_Bank |
| 2 Credit risk_risk_Assessment | 31 Agriculture_Rural_Revitalization |
| 3 Supply Chain_Logistics_Enterprise | 32 Construction_Construction Industry_Construction Industry |
| 4 Commercial Banks_Banking_Business | 33 Pattern_Dual Circulation_Cross-Strait |
| 5 Digital_Digitalization_Industry | 34 Food_Feed_Supermarket |
| 6 Competition_Supply Chain_Enterprise | 35 China_Industry Chain_Export |
| 7 SME_Financing_Information | 36 Rural_Village_Revitalization |
| 8 SMEs_Financing_Economy | 37 Ecological system_Ecosphere_Ecology |
| 9 China_SME_Innovation | 38 Bills_Industry Chain_CIRC |
| 10 Small_Medium_Enterprises | 39 Real Estate_Asset_Securitization |
| 11 Agriculture_Farmers_Agriculture | 40 Logistics_Enterprise_SME |
| 12 Agriculture_Agricultural products_Agriculture-related | 41 Agriculture_Farmers_Poverty |
| 13 Supply Chain Finance_Credit_Core | 42 Index_zx_China |
| 14 SMEs_Constraints_Financing | 43 Small_Enterprises_Medium |
| 15 SCF_Research_Proposed | 44 Rural Areas_Agriculture_Heilongjiang Province |
| 16 Supply Chain_Enterprise_SCF | 45 jt_Inner Mongolia_Region |
| 17 SMEs_Financing_Timber | 46 Project Management_Online_Maturity |
| 18 Constraints_Relief_Cash | 47 Logistics_Logistics Industry_Finance |
| 19 Agriculture_Digital_Rural | 48 Logistics_Third Party_Incentive |
| 20 Enterprise_Steel_Bank | 49 Research_Literature_Hotspots |
| 21 Automobile_New Energy_Automotive Industry | 50 Green_Low Carbon_Dual Carbon |
| 22 Pilot Zone_Free Trade Zone_Dongying City | 51 Accounts Receivable_Gold Order_Blockchain |
| 23 Data_Informatization_Wisdom | 52 Port_Ningbo_Distant-sea fishery |
| 24 Efficiency_Significant_Enterprise | 53 p2p_lending_online lending |
| 25 Green_finance_transformation | 54 Medicine_Circulation_Medical Devices |
| 26 YTO_Online_Supply Chain Finance | 55 Branch_Industry_Finance |
| 27 Reform_City_Supply | 56 Promote_Guide_Active |
| 28 Fraud_SCF_DiChain | |

central-level SCF policy theme Topic-1, which focuses on leveraging digital innovations in SCF to support agricultural production, empower rural revitalization, and alleviate poverty.

(3) The seven topics with serial numbers "52-33-22-54-51-20-4" are associated with keywords such as "banks" and industries involved in SCF, including pharmaceuticals, iron and steel, and offshore fishing. This section emphasizes how SCF outperforms traditional financing in supporting rural revitalization. It also provides recommendations for advancing SCF within the framework of China's dual-cycle development paradigm.

(4) The seven themes with serial numbers "48-40-47-15-37-49-26" revolve around keywords like "logistics," "ecology," and "SMEs." This segment focuses on improving logistics systems and developing SCF-related ecosystems, encompassing both the financial environment within supply chains and the sustainable development of ecological systems.

(5) The seven themes identified by the serial numbers "35-6-42-46-30-38-56" center on the keyword "industrial chain," with secondary keywords including "export," "competition," "online," "bills," "project management," and "maturity." This section highlights the digitization of China's industrial chain and the advancement of online business, advocating for the development of SCF and digital supply chain systems to address intensifying export competition.

(6) The five topics with serial numbers "1-23-5-0-53" include keywords such as "digitalization," "online," "technology," and "blockchain." These themes align with policy topic Topic-2, emphasizing the advancement of data and information technologies, particularly blockchain, to enhance SCF capabilities.

(7) The remaining 15 topics are more dispersed, covering a broader range of themes. For example, topics "25-50" explore green and low-carbon finance; "24-18" address SCF's role in alleviating enterprise capital constraints and improving efficiency; and "28-2" focuses on credit risk management among SCF participants.

## PMC-Index module construction

**Variable identification and indicator selection.** To reduce subjectivity, we developed the SCF policy evaluation index system. This system includes 10 primary variable indicators, such as policy nature ($X_1$), policy timeliness ($X_2$), policy subject ($X_3$), policy object ($X_4$), policy tool ($X_5$), policy attributes ($X_6$), policy objective ($X_7$), policy evaluation ($X_8$), policy scope ($X_9$), and effectiveness level ($X_{10}$). Additionally, it includes 41 secondary variables. Among them, the secondary variables under the four primary variables $X_4$, $X_5$, $X_6$, and $X_7$ are selected based on a combination of SCF business characteristics and the way SCF policy works. Table 4 displays the configuration of the evaluation indicators.

**Multi-input-output table construction.** Multi-input-output scales provide an alternative framework for data analysis that can store a large amount of data and calculate any of the individual variables. We create a multi-input-output scale to calculate 10 primary variables, each of which includes multiple secondary variables, without any quantitative limit on the number of secondary variables we can set. The weights assigned to each of the secondary variables are the same, so there is no need to rank them according to their degree of importance. Table 5 displays the multi-input-output scale of the SCF policy that this study established.

**PMC index calculation.** We utilized the SCF policy evaluation table to determine the correlation between primary and secondary variables. Using Eqs. (1) and (2), we calculated the values of secondary variables within the range [0, 1]. The arithmetic mean was employed to represent the policy value under a specific primary variable, calculated as the sum of the scores of all relevant secondary variables divided by their total number. Notably, the values of secondary variables are strictly confined to the [0, 1] range. Subsequently, based on Eq. (3), we computed the primary variable values for the SCF policy, enabling a multi-level policy assessment and the generation of corresponding numerical results. Finally, as shown in Eq. (4), the PMC index of the policy was derived by summing the values of all primary variables. This index allowed us to evaluate the SCF development policy and assign it a rating.

Given the limited number of relevant evaluation studies on SCF policies and the diverse attributes (e.g., economy, industry, and science and technology) encompassed by the selected and potential samples in this study, the evaluation criteria were adapted from similar studies [55]. Specifically, policies were classified as failing (score: [0, 3.99]), passing ([4, 5.49]), good ([5.5, 6.99]), or excellent (≥7.00).

**Table 4. SCF Policy Evaluation Indicator System.**

| Primary variables | Secondary variables | Evaluation basis for secondary variables (yes assign 1, no assign 0) |
|---|---|---|
| Policy nature ($X_1$) | Orientation ($X_{1-1}$) | Whether the policy has orientation properties |
| | Regulation ($X_{1-2}$) | Whether the policy has regulatory attributes |
| | Recommendation ($X_{1-3}$) | Whether the policy has the attribute of recommendation |
| | Description ($X_{1-4}$) | Whether the policy has descriptive properties |
| Policy timeliness ($X_2$) | Short-term ($X_{2-1}$) | Whether the policy has a statute of limitations of no more than 5 years |
| | Medium and long term ($X_{2-2}$) | Whether the policy is more than 5 years old |
| Policy issuing department (X3) | Single ($X_{3-1}$) | Whether the policy is issued by a single department |
| | Cooperation ($X_3 2$) | Whether the policy is issued jointly by multiple departments |
| Policy object ($X_4$) | Core Business ($X_{4-1}$) | Whether the policy content is targeted at core enterprises |
| | Service Platform ($X_{4-2}$) | Whether the policy is targeted at service platforms |
| | Regulatory Agency ($X_{4-3}$) | Whether the policy content targets regulators |
| | Financial Institutions ($X_{4-4}$) | Whether the policy content is targeted at financial institutions |
| | Government Departments ($X_{4-5}$) | Whether the policy content targets government departments |
| Policy tool ($X_5$) | Technical Support ($X_{5-1}$) | Whether the policy instrument involves technical support |
| | Talent Cultivation ($X_{5-2}$) | Whether the policy tool involves human resource development |
| | Information Service ($X_{5-3}$) | Whether the policy tool involves information services |
| | Research Projects ($X_{5-4}$) | Whether the policy tool involves research projects |
| | Regulatory Control ($X_{5-5}$) | Whether the policy tool involves regulatory control |
| | Risk Assurance ($X_{5-6}$) | Whether the policy tool involves risk protection |
| | Goal Planning ($X_{5-7}$) | Whether the policy tool involves target planning |
| | Behavioral Regulation ($X_{5-8}$) | Whether the policy tool involves behavioral regulation |
| | Scenario Construction ($X_{5-9}$) | Whether the policy tool involves scenario building |
| | Pilot Demonstration ($X_{5-10}$) | Whether the policy tool involves pilot demonstration |
| | Application Promotion ($X_{5-11}$) | Whether the policy tool involves application promotion |
| Policy attributes ($X_6$) | Finance ($X_{6-1}$) | Whether the policy contains financial attributes |
| | Technology ($X_{6-2}$) | Whether the policy contains technological attributes |
| | Supply Chain ($X_{6-3}$) | Whether the policy contains supply chain attributes |
| | System ($X_{6-4}$) | Whether the policy contains institutional attributes |
| Policy targets ($X_7$) | Broaden SCF Business Breadth ($X_{7-1}$) | Whether the policy content broadens the breadth of SCF business |
| | Enhance SCF Business Stability ($X_{7-2}$) | Whether the policy content enhances the stability of SCF business |
| | Deepen SCF business depth ($X_{7-3}$) | Whether the content of the policy deepens the depth of SCF business |
| Policy evaluation ($X_8$) | Sufficient basis ($X_{8-1}$) | Whether the policy basis is sufficient |
| | Clear objectives ($X_{8-2}$) | Whether the policy objective is clear |
| | Scientific program ($X_{8-3}$) | Whether the policy program is scientific |
| Policy scope ($X_9$) | National ($X_{9-1}$) | Whether the policy scope covers the whole country |
| | Regional ($X_{9-2}$) | Whether the policy scope covers a certain region |
| | Provincial and Municipal ($X_{9-3}$) | Whether the scope of the policy covers a province or city |
| Policy type ($X_{10}$) | Laws ($X_{10-1}$) | Type of policy is a law |
| | Working Papers ($X_{10-2}$) | Type of policy is a working document |
| | Regulations ($X_{10-3}$) | Type of policy is a regulation |
| | Normative Documents ($X_{10-4}$) | Policy type is a normative document |

**Table 5. Multi-input-output table for SCF policy evaluation.**

| Primary variables | $X_1$ | $X_2$ | $X_3$ | $X_4$ | $X_5$ | $X_6$ | $X_7$ | $X_8$ | $X_9$ | $X_{10}$ |
|---|---|---|---|---|---|---|---|---|---|---|
| Secondary variables | $X_{1\text{-}1}$ | $X_{2\text{-}1}$ | $X_{3\text{-}1}$ | $X_{4\text{-}1}$ | $X_{5\text{-}1}$ | $X_{6\text{-}1}$ | $X_{7\text{-}1}$ | $X_{8\text{-}1}$ | $X_{9\text{-}1}$ | $X_{10\text{-}1}$ |
| | $X_{1\text{-}2}$ | $X_{2\text{-}2}$ | $X_{3\text{-}2}$ | $X_{4\text{-}2}$ | $X_{5\text{-}2}$ | $X_{6\text{-}2}$ | $X_{7\text{-}2}$ | $X_{8\text{-}2}$ | $X_{9\text{-}2}$ | $X_{10\text{-}2}$ |
| | $X_{1\text{-}3}$ | | | $X_{4\text{-}3}$ | $X_{5\text{-}3}$ | $X_{6\text{-}3}$ | $X_{7\text{-}3}$ | $X_{8\text{-}3}$ | $X_{9\text{-}3}$ | $X_{10\text{-}3}$ |
| | $X_{1\text{-}4}$ | | | $X_{4\text{-}4}$ | $X_{5\text{-}4}$ | $X_{6\text{-}4}$ | | | | $X_{10\text{-}4}$ |
| | | | | $X_{4\text{-}5}$ | $X_{5\text{-}5}$ | | | | | |
| | | | | | $X_{5\text{-}6}$ | | | | | |
| | | | | | $X_{5\text{-}7}$ | | | | | |
| | | | | | $X_{5\text{-}8}$ | | | | | |
| | | | | | $X_{5\text{-}9}$ | | | | | |
| | | | | | $X_{5\text{-}10}$ | | | | | |
| | | | | | $X_{5\text{-}11}$ | | | | | |

$$X \sim [0,1] \tag{1}$$

$$X = \left\{ XR: [0 \sim 1] \right\} \tag{2}$$

$$X_i \left( \sum_{j=1}^{n} \frac{X_{i-j}}{T(X_{i-j})} \right) \tag{3}$$

where i (i = 1, 2, 3,..., m) represents the primary variable, and j (j = 1, 2, 3,..., n) represents the secondary variable.

$$PMC = \sum_{i=1}^{m} \left[ X_i \left( \sum_{j=1}^{n} \frac{X_{i-j}}{T(X_{i-j})} \right) \right] \tag{4}$$

**PMC surface construction.** To visualize PMC index results, the primary variable index scores for each policy were transformed into a 3rd-order matrix. By analyzing the concavity and convexity of the surface, we assessed the internal consistency of each policy: convex surfaces indicated higher scores for the corresponding indicators, while concave surfaces indicated lower scores. This graphical approach facilitated the evaluation of policy performance across different indicators. For enhanced clarity and balance, variable X10—contributing equally to each policy sample but consistently receiving lower scores—was excluded, leaving nine variables to construct the PMC surface. The plotting process is detailed in Eq. (5).

$$PMC\ surface = \begin{pmatrix} X_1 & X_2 & X_3 \\ X_4 & X_5 & X_6 \\ X_7 & X_8 & X_9 \end{pmatrix} \tag{5}$$

## Empirical analysis

### Selection of policy sample

To ensure the accuracy and validity of the policy samples, this study excludes policy texts that are no longer valid, have duplicated contents, or have low relevance to SCF, so that the

selected samples have high relevance. We divided the 181 policy texts into three stages of SCF development, considering the reasonableness of the distribution of policy samples, and selected the three policy texts with the highest degree of relevance to the theme of SCF development as research samples in each stage. The selected research samples are shown in Table 6, with $P_1$-$P_3$ representing Stage-1 policy samples, $P_4$-$P_6$ representing Stage-2 policy samples, and $P_7$-$P_9$ representing Stage-3 policy samples.

## SCF policy PMC-indexes

After comprehensively analyzing the policy themes and policy subwords generated based on the BERTopic model, we systematically assigned secondary variables in the multi-input-output table corresponding to the nine SCF policies and calculated the PMC index, policy samples' grades, and rankings as shown in Table 7 using Eqs. (1)–(4).

The results show that among the selected representative SCF policies, there are three policy samples with the rank of "good," six policy samples with the rank of "passing," and no policy samples with the rank of "failing" or "excellent." There are no policies with a grade of "fail" or "excellent." In terms of the formulation level of representative policy samples, the average value of the PMC index of policy samples ($P_7$–$P_9$) chosen by Stage-3 is 5.91, the average value of the PMC index of policy samples ($P_1$–$P_3$) chosen by Stage-1 is 5.1, and the average value of the PMC index of policy samples of Stage-2 ($P_4$–$P_6$) is only 4.74. As a result, it can be found that Stage-3 policymaking considerations are more comprehensive, while Stage-2 policy concerns are limited.

In addition, we plotted a Debra diagram to visualize the distribution of PMC indices for each level of variables, as shown in Fig 4. Out of the 10 variables, policy evaluation ($X_8$) has the highest mean value of 0.78. This means that China's SCF policies are well-thought-out and have a favorable mix of programs and methods. On the other hand, the variable policy tools ($X_5$) have the lowest mean value of 0.31, which means that China's SCF policies don't have many tools to help SCF grow and can still be improved. The PMC indices for the other Primary variables are all greater than or equal to the mean value. However, the PMC indices of policy attributes ($X_6$) and policy objectives ($X_7$) change more, indicating that policymaking

**Table 6. Sample PMC evaluation studies.**

| Number | Policy Title | Issue Date |
|---|---|---|
| $P_1$ | Notice on the Issuance of the Development Plan for the Integration of Informatization and Industrialization (2016-2020) | 2016.10.12 |
| $P_2$ | Notice on Doing a Good Job in Replicating and Extending the Comprehensive Reform Pilot Experiences of Supply and Marketing Cooperatives | 2017.04.22 |
| $P_3$ | Guiding Opinions on Actively Promoting Supply Chain Innovation and Application | 2017.10.05 |
| $P_4$ | Notice on Pilot Supply Chain Innovation and Application | 2018.04.10 |
| $P_5$ | 2019 China Inclusive Finance Development Report | 2019.09.29 |
| $P_6$ | Implementation Opinions on Promoting the Deep Integration and Development of Advanced Manufacturing and Modern Service Industries | 2019.11.10 |
| $P_7$ | Opinions on Regulating the Development of SCF to Support the Stable Circulation and Optimization and Upgrading of Supply Chain Industry Chain | 2020.09.18 |
| $P_8$ | Circular on the Issuance of Chengdu-Chongqing Plan for Building the Western Financial Center Together | 2021.12.13 |
| $P_9$ | Implementation Opinions on Promoting the High-Quality Development of Inclusive Finance | 2023.09.25 |

**Table 7. Representative SCF Policy PMC Index and Rating.**

| Primary variables | $P_1$ | $P_2$ | $P_3$ | $P_4$ | $P_5$ | $P_6$ | $P_7$ | $P_8$ | $P_9$ |
|---|---|---|---|---|---|---|---|---|---|
| $X_1$ | 0.5 | 0.5 | 1 | 0.75 | 0.75 | 0.75 | 1 | 0.5 | 1 |
| $X_2$ | 0.5 | 0.5 | 0.5 | 0.5 | 0.5 | 0.5 | 0.5 | 0.5 | 0.5 |
| $X_3$ | 0.5 | 0.5 | 0.5 | 0.5 | 0.5 | 0.5 | 0.5 | 0.5 | 0.5 |
| $X_4$ | 0.6 | 0.4 | 0.4 | 1 | 0.4 | 0.6 | 0.8 | 0.4 | 0.6 |
| $X_5$ | 0.18 | 0.18 | 0.55 | 0.27 | 0.18 | 0.18 | 0.45 | 0.36 | 0.45 |
| $X_6$ | 0.75 | 0.5 | 1 | 0.5 | 0.25 | 0.5 | 1 | 1 | 0.75 |
| $X_7$ | 0.33 | 0.33 | 0.67 | 0.33 | 0.67 | 0.67 | 0.67 | 0.33 | 1 |
| $X_8$ | 0.67 | 1 | 1 | 0.67 | 0.33 | 0.67 | 1 | 1 | 0.67 |
| $X_9$ | 0.33 | 0.33 | 0.33 | 0.33 | 0.33 | 0.33 | 0.33 | 0.33 | 0.33 |
| $X_{10}$ | 0.25 | 0.25 | 0.25 | 0.25 | 0.25 | 0.25 | 0.25 | 0.25 | 0.25 |
| PMC Index | 4.61 | 4.49 | 6.2 | 5.1 | 4.16 | 4.95 | 6.5 | 5.17 | 6.05 |
| Grade | passing | passing | good | passing | passing | passing | good | passing | good |
| Rank | 7 | 8 | 2 | 5 | 9 | 6 | 1 | 4 | 3 |

has evolved over time when it comes to choosing different perspectives on SCF development and setting different goals for SCF development.

## SCF policy PMC-surface

Overall, China has enacted and implemented a wide range of policy texts to promote SCF development, gradually enriching the variety of policy objectives and tools. By constructing a PMC surface diagram, this section provides a comprehensive assessment of the stability and shortcomings of a representative sample of policies at different stages of SCF development and proposes optimization paths for the corresponding policies.

**Stage-1 PMC surface analysis.** Fig 5 shows the PMC surface plotting for the Stage-1 policy sample $P_1$-$P_3$, where a single sector primarily formulates the policies with high correlation with the SCF in this period. A thorough analysis of the PMC surface plot reveals the following:

The P1 policy sample's PMC index is 4.61, ranking 7th with a passing evaluation grade. This policy, which was developed and promulgated by the Ministry of Industry and Information Technology (MIIT), aims to accelerate the construction of a strong manufacturing country and promote the deep integration of informationization and industrialization. However, this policy solely utilizes SCF as a financial tool to foster e-commerce and facilitate the integration and growth of manufacturing and the internet. The types of policy tools used are relatively homogeneous, focusing on application promotion and strengthening of information services. The PMC scores of this policy fall below the average for the Primary variables, which include policy nature ($X_1$), policy tools ($X_5$), policy objectives ($X_7$), and policy evaluation ($X_8$). Therefore, the primary focus for policy improvement should be on improving the aforementioned aspects, with the order of improvement being $X_1$-$X_7$-$X_5$-$X_8$.

The $P_2$ policy sample's PMC index is 4.49, ranking 8th with a passing evaluation grade. The item comes in last in the Stage-1 policy sample. The National Federation of Supply and Marketing Cooperatives (NFMC) issues the policy, and $P_2$ introduces SCF as a means of expanding the agricultural business to realize the conceptual leap from "earning money for farmers" to "helping farmers make money" to "making money with farmers." $P_2$ introduced SCF as a financial tool to expand agribusiness services for farmers and shared its experience. However, during this period, SCF has not yet become a mainstream financing tool for empowering agricultural production. Therefore, in this policy, the scores of the Primary variables, such as

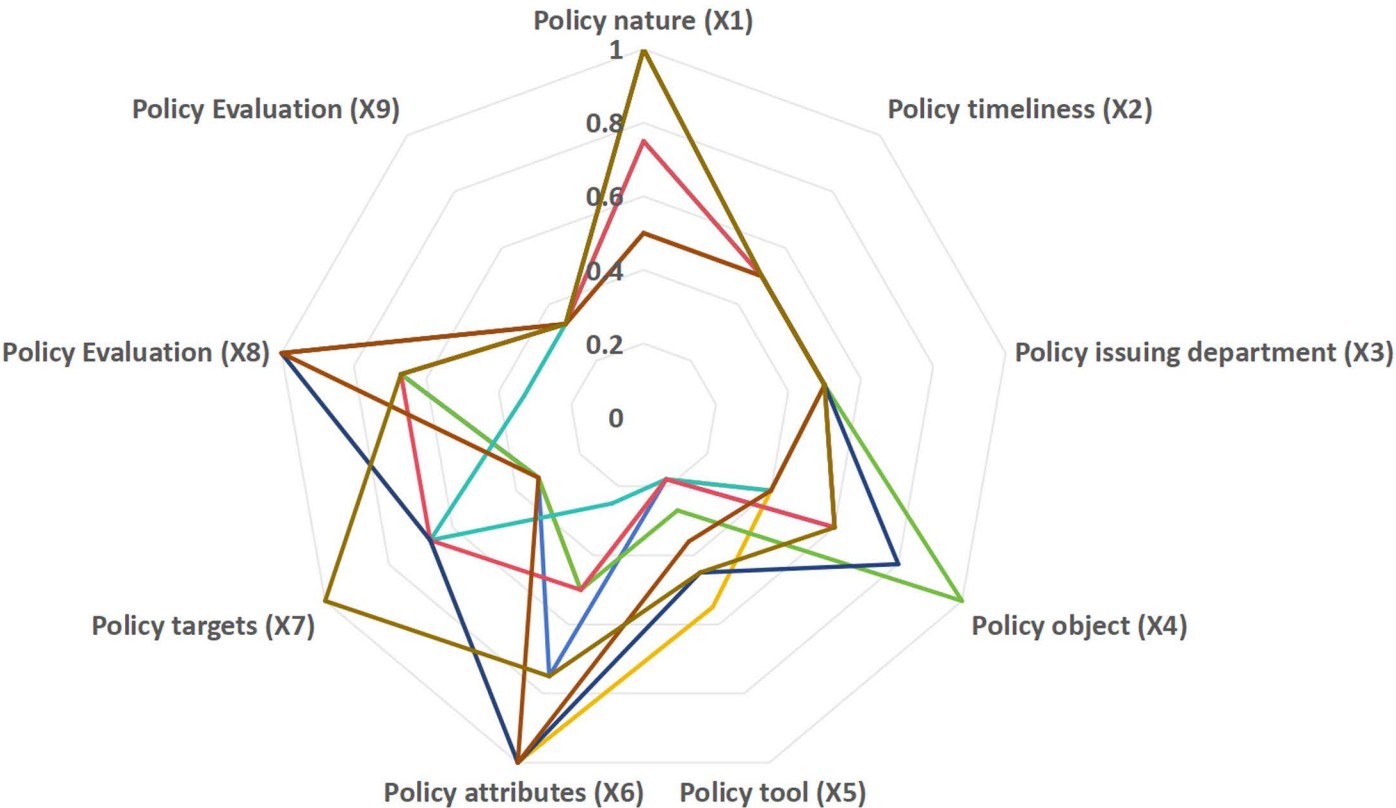

**Fig 4. Representative SCF Policy Debra Chart.**

policy nature ($X_1$), policy object ($X_4$), policy instrument ($X_5$), policy attributes ($X_6$), and policy evaluation ($X_7$), are lower than the average, whereas the scores of the other indicators are either higher or equal to the average. Therefore, when improving policies, it is necessary to consider the specific policy situation comprehensively, with the order of improvement being $X_1$-$X_7$-$X_6$-$X_4$-$X_5$.

The PMC index of the $P_3$ policy sample is 6.2, ranking No. 2 with a good evaluation grade. The General Office of the State Council promulgates the policy and holds high authority. Only one indicator, the policy object ($X_4$), is slightly lower than the average, with a difference of 0.18. The other indicators' scores are higher than or equal to the average, with a score of 0.55 for the policy tool ($X_5$), which is at the top of all the samples. This indicates that the policy is more reasonable in terms of design, and that the policy instruments are more comprehensive.

**Stage-2 PMC surface analysis.** Fig 6 plots the PMC surface of Stage-2 policy samples $P_4$-$P_6$, a stage where multiple departments primarily issue policies jointly, encompassing various fields of SCF innovation and development.

The $P_4$ policy sample's PMC index is 5.1, ranking 5th among all samples with a passing evaluation grade. Eight departments, including the Ministry of Commerce, the Ministry of

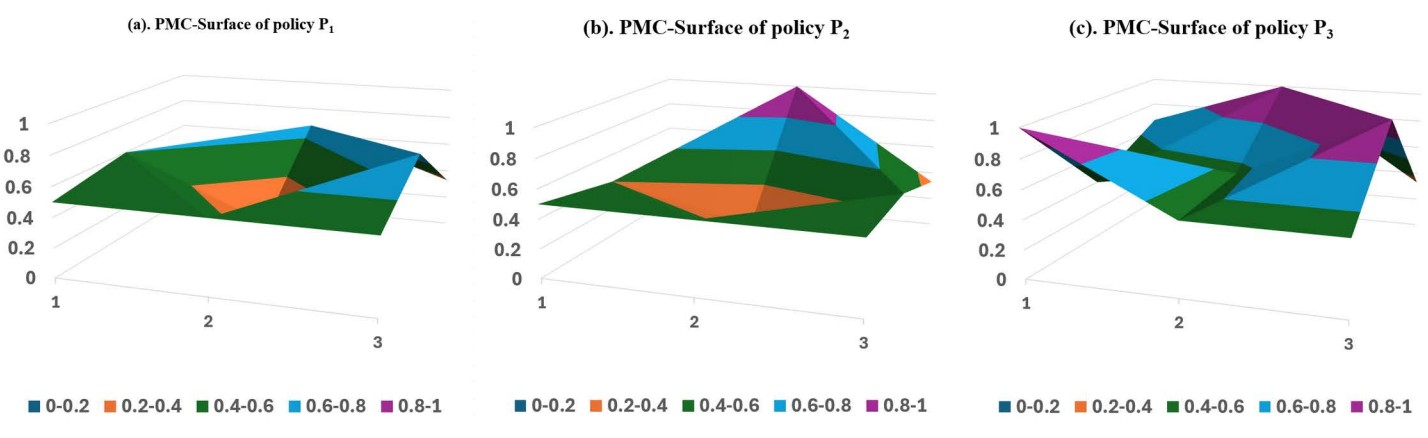

**Fig 5. PMC Surface of Stage-1 Policy Sample.**

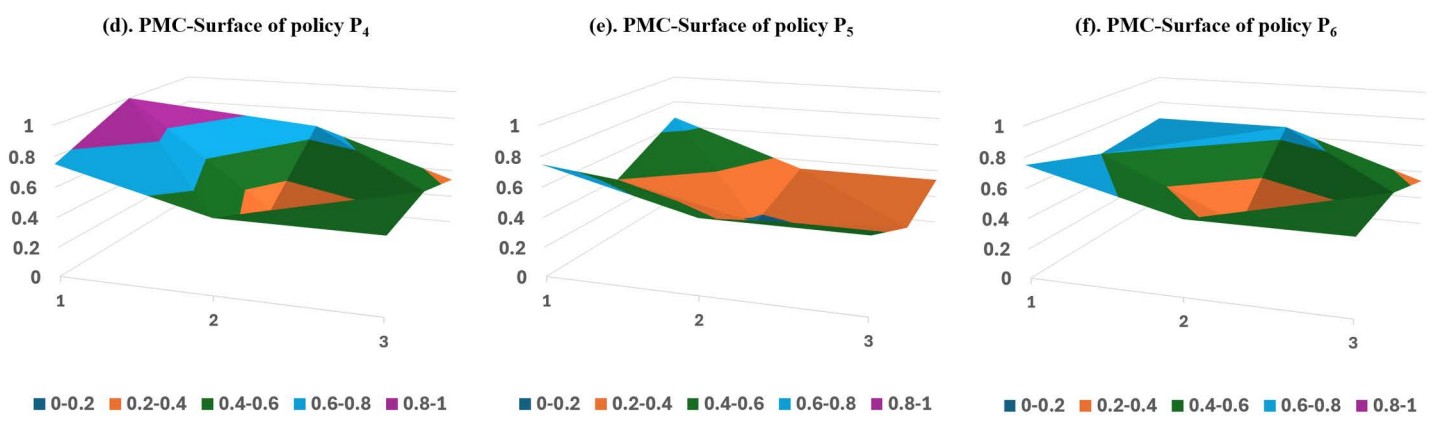

**Fig 6. PMC Surface of Stage-2 Policy Sample.**

Industry and Information Technology, and the Ministry of Ecology and Environment, jointly issue the policy. They have consistently emphasized the need to regulate the development of SCF to benefit the real economy. However, they have limited the policy tools to target planning and risk protection, without establishing specific milestones for SCF development. The scores on the indicators of the Primary variables policy tools ($X_5$), policy attributes ($X_6$), policy objectives ($X_7$), and policy evaluation ($X_8$) are all below average. The scores of policy instruments are slightly lower than the mean value, while the scores of policy attributes and policy evaluation are lower than the sample mean value of 0.19 and 0.11, respectively. Therefore, the order of improvement for this policy could be $X_7$-$X_6$-$X_8$-$X_5$.

The $P_5$ policy sample has a PMC index of 4.16, placing it 9th out of the total sample and earning a passing evaluation grade. The policy, jointly issued by the China Banking and Insurance Regulatory Commission and the People's Bank of China, aims to promote the deepening of structural reform on the financial supply side and proposes expanding the pilot scope of products such as SCF. The scores on the indicators for the Primary variables policy object ($X_4$), policy tools ($X_5$), policy attributes ($X_6$), and policy evaluation ($X_8$) are all below average. The PMC index for policy evaluation is below the mean value of 0.45, the PMC index for policy attributes is below the mean value of 0.44, and the scores of the other indicators are

above or equal to the mean value. Therefore, we can improve the policy in the following order: $X_8$-$X_6$-$X_4$-$X_5$.

The $P_6$ policy sample's PMC index is 4.95, ranking 6th with a passing evaluation grade. 15 ministries and commissions, including the National Development and Reform Commission, jointly issue the policy, demonstrating its strong authority. It primarily encourages the development of SCF by planning development goals and promoting its implementation. The scores on the indicators of the Primary variables, namely policy tools ($X_5$), policy attributes ($X_6$), and policy evaluation ($X_8$), are below average. Therefore, we can improve this policy in the order of $X_6$-$X_5$-$X_8$, but the improvement program should also take the specific policy situation into account.

**Stage-3 PMC surface analysis.** Entering Stage 3, the SCF chaos in China is under control, and its development is gradually on track. Fig 7 plots the PMC surfaces of policy samples $P_7$–$P_9$. Multiple ministries and commissions jointly formulate the majority of the policies at this stage, and they have enriched and standardized the tools for optimizing SCF application.

The P7 policy sample's PMC index is 6.5, ranking first among all nine SCF development policies with a good evaluation rating. The People's Bank of China, the Ministry of Industry and Information Technology, and other 8 ministries and commissions jointly issue this policy, which provides a series of guidelines on the standardized development of SCF, including risk protection, target planning, and technical support. These guidelines mark a new stage in the development of China's SCF. The PMC indices of all Primary variables in this policy are higher than or equal to the average value, indicating that the policy was formulated with a more comprehensive consideration of various factors.

The PMC index of the P8 policy sample is 5.17, ranking 4th among all the samples with a passing evaluation grade. Eight ministries and commissions, including the People's Bank of China and the National Development and Reform Commission, jointly issued the policy. In order to promote the construction of the twin-city economic circle in the Chengdu-Chongqing region, this plan aims to improve the financial service system, including SCF, and proposes to promote the innovative development of SCF and trade finance. The scores on the Primary variables policy nature ($X_1$), policy object ($X_4$), and policy objective ($X_7$) are lower than the average value, while the scores on the other indicators are higher than or equal to the average value. Therefore, we can prioritize improving these two aspects of the policy, following the specific order of $X_1$-$X_7$-$X_4$.

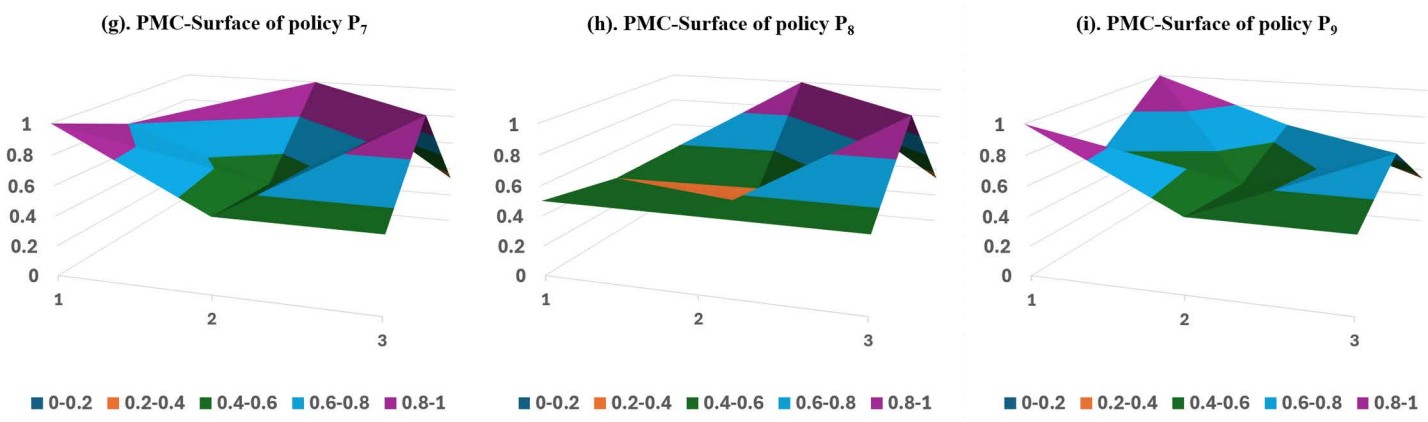

**(g). PMC-Surface of policy $P_7$**   **(h). PMC-Surface of policy $P_8$**   **(i). PMC-Surface of policy $P_9$**

**Fig 7. PMC Surface of Stage-3 Policy Sample.**

The $P_9$ policy sample's PMC index is 6.05, ranking No. 3, with an evaluation grade of good. The State Council promulgated the policy to establish an inclusive financial system at a high level, aiding in the successful execution of the national rural revitalization strategy. $P_9$ views agricultural SCF as a valuable instrument for a robust rural financial service system. The policy is only slightly below average in one indicator of policy evaluation ($X_8$), with a difference of 0.11. The scores of other indicators are either higher than or equal to the average, indicating that the policy is more reasonable in terms of setting policy tools and objectives, and its attributes are more comprehensive, with an optimal path of $X_8$.

## Conclusions and implications

### Conclusions

This study collects 181 Chinese central-level SCF policies released as of January 2024 and 3,439 research abstracts of Chinese papers focusing on SCF development, conducts thematic cluster analysis based on the BERTopic model, and combines the PMC-Index model to quantitatively evaluate nine representative SCF policies. Upon quantitative evaluation, the study reaches the following conclusions:

(1) Policy themes are more limited. Policies and scientific research are equally important factors for SCF development. China's SCF research currently encompasses a wide range of themes, including promoting digitization in various fields like commerce and agriculture and managing financial risks. However, the national SCF-related policy provisions heavily focus on SCF model innovation and risk management in supply chain SMEs financing.

(2) There are fluctuations in the level of policy formulation. From Stage-1 to Stage-2 to Stage-3, the PMC indexes of representative SCF policies are 5.1, 4.74, and 5.91, respectively. This is because in Stage 2, the scores of the three first-level indexes, namely policy tools ($X_5$), policy attributes ($X_6$), and policy evaluation ($X_8$), are low, but the scores of the above indexes increase rapidly when entering Stage-3.

(3) Homogenization of policy content is obvious. Examining the specific distribution of the PMC index of SCF's representative policies reveals a relatively concentrated score distribution for the secondary variables under the two first-level indicators of policy object ($X_4$) and policy instrument ($X_5$). Specifically, policy targets ($X_4$) are mostly government departments ($X_{4-5}$) and financial institutions ($X_{4-4}$), while the rest of the types of SCF participants are less likely to be the main targets of policies. Furthermore, information services ($X_{5-3}$), risk protection ($X_{5-6}$), target planning ($X_{5-7}$), and application promotion ($X_{5-11}$) dominate the use of policy tools ($X_5$), demonstrating a relatively homogeneous use of these tools.

### Policy recommendations

Based on the PMC assessment results of SCF policy, this study combines the clustering analysis of China's SCF scientific research themes and puts forward suggestions for improving and revising China's SCF policy.

(1) Expanding policy focus: SCF empowers supply chain SMEs to improve information flow efficiency and access to capital. Future policy focus should be on the implementation of the SCF business model, with an emphasis on the optimization of risk management mechanisms, the construction of green financial systems, and the application of digital technology innovation.

(2) Different types of policies should form a synergy. Policy factors highly influence the development trajectory of SCF; regulatory policies and policies on the use and

promotion of SCF should be complementary to each other; subsequent policy development should be more systematic and comprehensive; and policies should form a synergy.

(3) Expand the types of policy tools used. Currently, policy tools primarily concentrate on promoting applications and planning targets, with the primary goal of encouraging banks and other financial institutions to implement innovative SCF applications. However, the development of SCF depends on optimizing the financial environment and multi-level government supply. The future development of SCF policy should increase the use of policy tools should increase as SCF policy develops.

(4) Diversified policy focus: The SCF policy focuses on widely distributed SMEs at all levels in the supply chain, but the main target of the current policy is banks and other financial institutions, and only a small part of the policy specifies the responsibilities and obligations of the core enterprises and service platforms in the supply chain. In the future, the policy should diversify its target and decentralize the allocation of risk management responsibilities.

## Research outlook

In this paper, we use the BERTopic model to sort out the policy and research themes and innovatively construct a SCF policy evaluation system based on the theme clustering results combined with the PMC-Index model, which provides specific and actionable suggestions for the subsequent development of SCF policies. Reasonable suggestions to promote the development of SCF are put forward from a policy perspective, providing new research ideas and theoretical foundations for the field of SCF research.

However, the research scope and time constraints constrain this study; the SCF policy's PMC evaluation index system lacks comprehensiveness, and the allocation process for each index remains subjective, necessitating further improvement in objectivity. At the same time, the text of SCF policy is sufficient, and the subject of the text is still being enriched, so future research can focus on how SCF policy affects the performance evaluation within the supply chain system to carry out studies.

## Supporting information

**Appendix. Table of variable assignments for representative SCF policy samples.** (DOCX)

## Author contributions

**Conceptualization:** Yin Dong.

**Data curation:** Mingyang Li.

**Methodology:** Mingyang Li.

**Software:** Mingyang Li.

**Supervision:** Yin Dong.

**Validation:** Yin Dong.

**Visualization:** Mingyang Li.

**Writing – original draft:** Mingyang Li.

**Writing – review & editing:** Mingyang Li.

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
