## [Decision Letter · Decision Letter 0]

19 Nov 2024

PONE-D-24-45592Research on the Evaluation of China's SCF Policy Based on Text MiningPLOS ONE

Dear Dr. Li,

Thank you for submitting your manuscript to PLOS ONE. After careful consideration, we feel that it has merit but does not fully meet PLOS ONE’s publication criteria as it currently stands. Therefore, we invite you to submit a revised version of the manuscript that addresses the points raised during the review process.

We look forward to receiving your revised manuscript.

Kind regards,

Mazyar Ghadiri Nejad, Ph.D.

Academic Editor

PLOS ONE

“National Social Science Fund of China

(Award Number: 23BTQ077).”

5. Please ensure that you include a title page within your main document. You should list all authors and all affiliations as per our author instructions and clearly indicate the corresponding author.

Reviewers' comments:

Reviewer's Responses to Questions

**Comments to the Author**

1. Is the manuscript technically sound, and do the data support the conclusions?

Reviewer #1: Yes

Reviewer #2: Yes

Reviewer #3: Yes

Reviewer #4: Yes

2. Has the statistical analysis been performed appropriately and rigorously? 

Reviewer #1: Yes

Reviewer #2: Yes

Reviewer #3: Yes

Reviewer #4: Yes

3. Have the authors made all data underlying the findings in their manuscript fully available?

Reviewer #1: No

Reviewer #2: Yes

Reviewer #3: No

Reviewer #4: Yes

4. Is the manuscript presented in an intelligible fashion and written in standard English?

Reviewer #1: Yes

Reviewer #2: Yes

Reviewer #3: Yes

Reviewer #4: Yes

5. Review Comments to the Author

Reviewer #1: Peer Review Report

1. Overall Evaluation

This paper addresses a novel and valuable topic, focusing on the evaluation of Supply Chain Finance (SCF) policies in China. By employing the BERTopic model and the PMC index model for text clustering and quantitative analysis, the paper provides an innovative and scientifically rigorous framework for policy assessment. The study’s methodology is sound and offers significant theoretical and practical value.

2. Strengths

Research Significance: The paper’s topic is highly relevant, aligned with the current SCF development context in China, and it contributes a fresh perspective to the evaluation of SCF policy effectiveness, offering valuable theoretical support for future policy optimization.

Methodological Innovation: The integration of the BERTopic and PMC index models in policy research is relatively uncommon and expands the methodological approaches available for SCF policy evaluation, offering a reference for similar studies.

3. Suggested Improvements

Title Clarity:It is recommended to avoid using abbreviations in the title, such as “SCF.” Writing out “Supply Chain Finance” will help readers quickly understand the subject of the study.

Abstract Enhancement:Consider expanding the abstract to more comprehensively reflect the core elements of the study, including the research purpose, methodology, findings, and conclusions. Including specific information, such as sample size, would further enrich the abstract and draw readers’ interest.

Enrichment of Literature Review and Theoretical Background: The methodology-related content currently included in the “Literature Review” section should be moved to the “Research Methodology” section to improve logical flow. Additionally, expanding the literature review to include a broader discussion of SCF policies, the BERTopic model, and the PMC index model will provide a stronger theoretical foundation for the analysis. To strengthen the literature review, consider adding more studies related to SCF policy development and theoretical background, particularly regarding the applications of the BERTopic model in policy analysis and the theoretical basis for using the PMC index model. This will provide stronger support for the research design and methodology choices.

4. Language and Expression

Certain sections could be expressed more concisely to improve readability. Consider revising sentence structure to enhance the overall quality of English expression.

Reviewer #2: Dear Author, Please, consider this comment

under the part of literature you need to define your technical terms like, text mining, supply chain finance, and cluster analysis so that the readers can clearly understand the context of your study.

Reviewer #3: having read through the manuscript, I have the conviction that authors did a good writeup which should be published. however, at this time, I am unable to write a full review comments which should have been very little by the way.

Reviewer #4: Study is well conceived. Literature review is up to the mark, methodology is good. Conclusions and discussions are at par with the objective of the study. Ethical approval seems missing, If authors can provide the details in methodology, it will be much better.

6. PLOS authors have the option to publish the peer review history of their article (what does this mean? ). If published, this will include your full peer review and any attached files.

**Do you want your identity to be public for this peer review?** For information about this choice, including consent withdrawal, please see our Privacy Policy .

Reviewer #1: **Yes: ** XINXIANG GAO

Reviewer #2: **Yes: ** Gudeta Shanko Charu

Reviewer #3: No

Reviewer #4: No

---

## [Author Response · Author response to Decision Letter 1]

26 Dec 2024

Dear Editor and Reviewers,

Thank you for your letter and for there viewers' comments concerning our manuscript entitled "Research on the Evaluation of China's SCF Policy Based on Text Mining" (ID: PONE-D-24-45592). Those comments are all valuable and very helpful for revising and improving our paper, as well as the important guiding significance to our research. We have studied comments carefully and have made corrections, which we hope meet with approval. Revised portions are marked in red underline in the paper.

The main corrections in the paper and the responses to the reviewer's comments are as flowing:

Responds to journal requirements:

1.Comment:

Response:

We apologize that our previous submission did not fully meet your journal's requirements. At present, we have carefully modified the article format according to the style template file you provided.

2.Comment:

Please ensure that you include a title page within your main document.

Response:

The revised manuscript has placed the title page at the beginning of the manuscript file.

3.Comment:

Please state what role the funders took in the study.

Response:

The second author's responsibilities for this study are: Conceptualization, Supervision, and Validation. In addition, after our confirmation, this study did not obtain any funding support, so we will modify the corresponding information in the revised submission.

4.Comment:

Recommend all authors decide on a data sharing plan before acceptance.

Response:

After discussion among all authors, we confirm acceptance of your journal's data sharing plan: all data will be freely accessible after the manuscript is accepted for publication. But we don't know what actions we should take to show that we support the plan.

5.Comment:

Please ensure that you include a title page within your main document.

Response:

The revised manuscript now lists all authors and all affiliations in accordance with your journal's instructions for authors and clearly indicates the corresponding author.

6.Comment:

Please review your reference list to ensure that it is complete and correct.

Response:

Thank you for your reminder. We have added 36 new references in the revised paper. In addition, we have carefully checked all the cited literature to ensure that there are no studies with "retracted" status among them.

At the end of this letter we provide detailed information on the changes to the references.

Responds to the reviewers' comments:

Reviewer #1:

1.Comment:

It is recommended to avoid using abbreviations in the title, such as "SCF" Writing out "Supply Chain Finance" will help readers quickly understand the subject of the study.

Response:

Your suggestion is very accurate and helpful, and we have changed the "SCF" in the title to "Supply Chain Finance".

2.Comment:

Consider expanding the abstract to more comprehensively reflect the core elements of the study, including the research purpose, methodology, findings, and conclusions. Including specific information, such as sample size, would further enrich the abstract and draw readers' interest.

Response:

According to your suggestion, we have revised the abstract. Specifically, we have adopted a more coherent writing logic and enriched the content of the research purpose and methodology with specific research information. You can see them in lines 21 to 29 of the revised manuscript.

3.Comment:

The methodology-related content currently included in the "Literature Review" section should be moved to the "Research Methodology" section to improve logical flow. Additionally, expanding the literature review to include a broader discussion of SCF policies, the BERTopic model, and the PMC index model will provide a stronger theoretical foundation for the analysis. To strengthen the literature review, consider adding more studies related to SCF policy development and theoretical background, particularly regarding the applications of the BERTopic model in policy analysis and the theoretical basis for using the PMC index model. This will provide stronger support for the research design and methodology choices.

Response:

According to your requirements and suggestions, we have made the following changes:

First, we adjusted the position and expression strategy of the contents related to computer technology in the literature review. In the revised manuscript, they are located in lines 209-229 as the "Main technologies for indicator selection" section. This section is used to explain how to achieve the technical requirements of this study, that is, to build a more optimized policy evaluation index system, the development trajectory of related technologies and their advantages and disadvantages, and why this study chooses to use the BERTopic model to optimize the construction of the index system.

Second, we revised the literature review section. Specifically, we added a review of SCF policy-related research (lines 85-105) and revised the review of policy evaluation methods (lines 134-172).In the "SCF Policy-Related Research" section, we analyzed the main directions and shortcomings of existing research, and introduced a broader discussion on combining technical means with SCF policy evaluation. In the "Policy Evaluation Methods" section, we systematically sorted out the existing policy evaluation methods, and introduced the PMC index model by comparing the evaluation performance of different models. Furthermore, based on the subjective problem of the PMC index model, we introduced the idea of using technical means to optimize the process of selecting policy evaluation indicators.

Thank you for your careful reading of our study and your professional and constructive comments. We believe that the revisions in this section more effectively support the research design and method selection.

4.Comment:

Certain sections could be expressed more concisely to improve readability. Consider revising sentence structure to enhance the overall quality of English expression.

Response:

According to your request, we have rechecked the academic norms of each paragraph and tried to present it in a more readable language. Specifically, the lines we have revised are as follows: 37-83, 245-255, 257-261, 271-274, 283-314, 335-349, 357-364.

Reviewer #2:

1.Comment:

Under the part of literature you need to define your technical terms like, text mining, supply chain finance, and cluster analysis so that the readers can clearly understand the context of your study.

Response:

Thank you for your sincere comments. We have added a "Concept definition" section between "Literature Review" and "Research Framework" (lines 173-178). This section introduces the definitions of the main terms used in the article and how they are interpreted in this study. In addition, we have plotted the results in a table to improve readability.

Reviewer #3:

Thank you for taking the time to read our research, and we're glad it received such high praise from you. We wish you all the best!

Reviewer #4:

1.Comment:

Ethical approval seems missing, If authors can provide the details in methodology, it will be much better.

Response:

Thank you for taking the time to read our research. The questions you raised are very important.

First of all, we attach great importance to the research norms regarding ethics. However, in this study, no living organisms were involved in the experiments, and all the data or texts used as research subjects can be publicly queried on the Internet. Therefore, we did not add the ethical approval section to the paper. Of course, if you think it is necessary to include this statement, we will consider it.

Second, regarding the details of the methodology, in the revised paper, we added "Main technologies for indicator selection" (lines 209-229), which, combined with the "Policy Evaluation Methods" in the literature review, will help readers better understand the methodology used in this study.

Reference revision list:

Due to the revision of some parts of the literature review, the references cited in the corresponding parts need to be reconsidered. Specifically, the articles numbered 1-5, 20-25, 27-29, 37, 40-41, and 43 in the original references were not replaced, and they were numbered 1-5, 20-25, 27-29, 49, 52-53, and 55 in the revised manuscript. In addition, in the revised paper, we added 36 new references, which are numbered 6-19, 26, 30-48, and 50-51.

Additional Notes：

This revision does not involve the main technical content of the research. Based on the professional opinions of the reviewers, we focused on improving the theoretical basis of the article and modified other details that can improve the readability of the research. In addition, in the revised article, all images are .tif format files modified using the PACE tool. Finally, when submitting, we will upload the images in the article and the text data used in the research in the form of a compressed file.

We appreciate the editors'/reviewers earnest work and hope that the correction will meet with approval.

Once again, thank you very much for your comments and suggestions.

Sincerely,

Corresponding author.

---

## [Decision Letter · Decision Letter 1]

5 Jan 2025

Research on the Evaluation of China's Supply Chain Finance Policy based on Text Mining

PONE-D-24-45592R1

Dear Dr. Sunny Li,

We’re pleased to inform you that your manuscript has been judged scientifically suitable for publication and will be formally accepted for publication once it meets all outstanding technical requirements.

Kind regards,

Mazyar Ghadiri Nejad, Ph.D.

Academic Editor

PLOS ONE

Reviewers' comments:

Reviewer's Responses to Questions

**Comments to the Author**

1. If the authors have adequately addressed your comments raised in a previous round of review and you feel that this manuscript is now acceptable for publication, you may indicate that here to bypass the “Comments to the Author” section, enter your conflict of interest statement in the “Confidential to Editor” section, and submit your "Accept" recommendation.

Reviewer #5: All comments have been addressed

2. Is the manuscript technically sound, and do the data support the conclusions?

Reviewer #5: Yes

3. Has the statistical analysis been performed appropriately and rigorously? 

Reviewer #5: Yes

4. Have the authors made all data underlying the findings in their manuscript fully available?

Reviewer #5: Yes

5. Is the manuscript presented in an intelligible fashion and written in standard English?

Reviewer #5: Yes

6. Review Comments to the Author

Reviewer #5: The manuscript is structured with clear sections; however, it could benefit from improved readability. Some sections are dense with information and may overwhelm readers unfamiliar with the topic. A more concise presentation of key findings and clearer transitions between sections would enhance overall clarity.)

7. PLOS authors have the option to publish the peer review history of their article (what does this mean? ). If published, this will include your full peer review and any attached files.

**Do you want your identity to be public for this peer review?** For information about this choice, including consent withdrawal, please see our Privacy Policy .

Reviewer #5: **Yes: ** Dr. AISHA BADAWI ABDELRHMAN

---

## [Editor Report · Acceptance letter]

PONE-D-24-45592R1

PLOS ONE

Dear Dr. Li,

I'm pleased to inform you that your manuscript has been deemed suitable for publication in PLOS ONE. Congratulations! Your manuscript is now being handed over to our production team.

Kind regards,

on behalf of

Assoc. Prof. Dr. Mazyar Ghadiri Nejad

Academic Editor

PLOS ONE